# Chemical Profile, Antimicrobial and Antioxidant Activity Assessment of the Crude Extract and Its Main Flavonoids from Tartary Buckwheat Sprouts

**DOI:** 10.3390/molecules27020374

**Published:** 2022-01-07

**Authors:** Lingyun Zhong, Yuji Lin, Can Wang, Bei Niu, Ying Xu, Gang Zhao, Jianglin Zhao

**Affiliations:** 1College of Medicine, Chengdu University, Chengdu 610106, China; zhongly@cdu.edu.cn (L.Z.); niubeicdu@163.com (B.N.); 2Key Laboratory of Coarse Cereal Processing, Ministry of Agriculture and Rural Affairs, Chengdu 610106, China; greenslevees@cdu.edu.cn (Y.L.); Wangcancdedu@163.com (C.W.); 040108222@163.com (Y.X.); ccpczhaogang@163.com (G.Z.)

**Keywords:** Tartary buckwheat sprouts, crude extract, flavonoids, bioactivity

## Abstract

The purpose of this study was to investigate the major flavonoids content and bioactivities of Tartary buckwheat sprouts. The crude methanol extract (ME) of Tartary buckwheat sprouts was abundant in flavonoids, and six major flavonoids, including isoorientin, vitexin, isovitexin, rutin, quercetin, and kaemferol were successfully determined from the sprouts by the high-performance liquid chromatography (HPLC) method. Generally, the flavonoid content of buckwheat sprouts was in the order of rutin > quercetin > isovitexin > vitexin> isoorientin > kaemferol. The highest rutin content of the ME and sprout cultures was 89.81 mg/g and 31.50 mg/g, respectively. Antibacterial activity results indicated the ME displayed notable inhibitory activity against the five tested bacteria, and its minimum inhibitory concentration (MIC) values ranged from 0.8 mg/mL to 3.2 mg/mL. Among the six flavonoids, quercetin was the most active compound, which exhibited strong activity against all tested bacteria except for *E. coli* and *S. epidermidis*, with its MIC values ranging from 0.2 mg/mL to 0.4 mg/mL. For the antifungal activity assay, the ME of Tartary buckwheat sprouts and four flavonoids could significantly inhibit the spore germination of two pathogenic fungi, and their inhibitory efficiency was concentration dependent. Quercetin was the most active one, which significantly inhibited the spore germination of *F. oxysporum* f. sp. *vasinfectum* and *F. oxysporum* f. sp. *cucumerinum*, and its median effective inhibitory concentration (IC_50_) value was 42.36 and 32.85 µg/mL, respectively. The antioxidant activity results showed that quercetin, kaemferol, and rutin displayed excellent antioxidant activity in the DPPH radical scavenging test, and their IC_50_ value was calculated as 5.60, 16.23, and 27.95 µg/mL, respectively. This is the first report on the antimicrobial activity of the crude extract of Tartary buckwheat sprouts. These results indicated that the methanol extract of Tartary buckwheat sprouts could be used as a potential antimicrobial or antioxidant agent in the future.

## 1. Introduction

*Fagopyrum tataricum* (L.) Gaertn. (Tartary buckwheat), of the Polygonaceae family, is a well-known edible and medicinal coarse cereal that is widely consumed around the world [1]. It is mainly cultivated in southwestern China, northern India, Nepal, and Bhutan, and has also been utilized as the staple food by minority people in southwestern China for a long time [2,3]. Tartary buckwheat has been recognized as an outstanding nutrition food, because of its abundant contents in proteins, amino acids, dietary fiber, vitamins, trace elements, and flavonoids [4,5,6]. In particular, flavonoids from Tartary buckwheat contributed various pharmacological effects that could benefit human health, mainly including notable antioxidant and anti-inflammatory activity, coronary heart disease prevention, hepatoprotective effects [7,8], etc., and this has attracted the interest of many researchers.

Tartary buckwheat sprout, developed from healthy seeds, is a kind of novel nutritional vegetable and is usually consumed as a salad in East Asian countries such as China, Japan, and Korea [9,10]. Buckwheat sprout cultures have a crisp texture, slightly sour and bitter taste, with an appealing fragrance. With their unique tasty flavor, buckwheat sprouts have gained much attention, both as gourmet food and as subjects of research, and they have become even more popular in recent years [11,12,13]. Buckwheat sprouts, which are considered edible sources of rutin, contain a greater abundance of flavonoids and other polyphenols than buckwheat seeds [14]. Additionally, the types or amounts of flavonoids can be significantly enhanced, and the nutritional quality and functionality of buckwheat have also been efficiently improved by sprouting [15]. The phytochemical profiles of buckwheat sprouts have been investigated for more than 10 years. However, the types and contents of flavonoids in buckwheat sprouts vary considerably, which is closely related to their varieties, malting stages, and cultivation conditions [16,17,18]. Some studies found that Tartary buckwheat contains much more phenolic components than that of the common buckwheat, which is another important cultivated species in the genus of *Fagopyrum* [19]. In common buckwheat sprouts, rutin, orientin, isoorientin, vitexin, and isovitexin have been detected most often [20,21]. It has been reported that Tartary buckwheat sprouts mainly contain rutin and quercetin, whereas other flavonoids (i.e., vitexin, isovitexin, kaemferol) are in trace amounts or have not even been detected in some varieties [22,23]. Therefore, it is insightful to investigate and confirm the main flavonoids in Tartary buckwheat sprouts for further developing their nutritional and medicinal values. In addition, Tartary buckwheat could considerably synthesize flavonoids during the germination and sprouting process, which may help the host to fight against the phytopathogenic microorganism infection, and these flavonoid compounds may have potential antimicrobial activities [24,25]. As far as we know, there is little information regarding the antimicrobial activity of Tartary buckwheat sprouts thus far, and there is an urging need to investigate and clarify the antimicrobial flavonoids from Tartary buckwheat sprouts. In this study, the methanol extract of Tartary buckwheat sprout cultures was prepared. Then, its major flavonoids were analyzed and quantified. Moreover, the antimicrobial and antioxidant activities of the crude methanol extract (ME) and its main flavonoids were also evaluated, and the structure–activity relationship (SAR) of the flavonoids was further elucidated.

## 2. Results and Discussion

### 2.1. Chemical Profile of the Crude Extract of Tartary Buckwheat Sprouts

The total flavonoids content of the crude methanol extract (ME) of Tartary buckwheat sprouts was measured, and the results indicated the ME was abundant in flavonoids, and its content was determined as 98.6 mg/g (the regress equation used for calculation of total flavonoids was Y = 0.0018X − 0.0227, R^2^ = 0.9998). The six main flavonoids including isoorientin, vitexin, isovitexin, rutin, quercetin, and kaemferol, were well separated from the buckwheat sprouts extract (presented in Figure 1B), and preliminarily identified by comparing the retention times with the standards.

As listed in Table 1, the rutin, which is a flavonol glycoside known to protect blood vessels, appeared as the most abundant flavonoid in the buckwheat sprout cultures. The rutin content was as much as 89.81 mg/g in the crude extract (ME), and 31.50 mg/g in Tartary buckwheat sprouts, respectively, followed by the quercetin content, which was 23.34 mg/g in the ME and 8.17 mg/g in the sprout cultures. Generally, the flavonoid content of the crude extract was in the order of rutin > quercetin > isovitexin > vitexin > isoorientin > kaemferol. The quantities of isoorientin and kaemferol were fairly low, and their concentrations in the crude extract of buckwheat sprouts were only 0.92 mg/g and 0.86 mg/g, respectively.

The flavonoid components of buckwheat sprouts have been investigated for more than 10 years, although the controversy about the flavonoid types of buckwheat sprouts still exists. It was reported that the main flavonoid of Tartary buckwheat sprouts was rutin, and the isoorientin or/and kaemferol could not be present in the sprout cultures [16,26]. Nevertheless, isoorientin, kaemferol, and the other four flavonoid compounds have been successfully detected in this investigation. The main reason could be due to the different Tartary buckwheat varieties used in the researchers’ tests, as the flavonoids constituent or content might be different in the host plant. In addition, another reason might be attributed to the fact that the flavonoids content of the crude methanol extract (ME) was much higher than that of the buckwheat sprout cultures, which made them easier to be detected by the HPLC analysis. In order to understand the preliminary biological activities of the main flavonoids present in Tartary buckwheat sprout cultures, their antimicrobial and antioxidant activities were evaluated in detail in this study.

### 2.2. Antimicrobial Activity

#### 2.2.1. Antibacterial Activity of the Buckwheat Sprout Extract and Its Main Flavonoids

As shown in Table 2, the crude methanol extract (ME) of Tartary buckwheat sprout cultures exhibited notable inhibitory activity against the five tested bacteria, including two Gram-negative strains (*P. lachrymans* and *S. typhimurium*) and three Gram-positive bacteria (*B. subtili*, *S. albus,* and *S. aureus*), and their MIC values ranged from 0.8 mg/mL to 3.2 mg/mL. Of the nine tested bacteria, *P. lachrymans* was the most sensitive bacterium to ME, and its MBC value was determined as 1.6 mg/mL.

As regards the antibacterial activity of the six major flavonoids, quercetin was the most active one, and it showed strong antibacterial activity toward all the tested bacteria except for *E. coli* and *S. epidermidis*. The MIC values of quercetin against *A. tumefaciens*, *X. vesicatoria*, *P. lachrymans* and *B. subtili* was determined as 0.2 mg/mL, 0.2 mg/mL, 0.4 mg/mL and 0.4 mg/mL, respectively. Correspondingly, the MBC values of quercetin toward *P. lachrymans* and *B. subtil* were determined to be 0.6 mg/mL and 0.6 mg/mL, respectively. Overall, the tested plant pathogenic bacteria were more sensitive than that of the clinic bacteria to quercetin. It could imply that quercetin may directly help the host plant or participate in the resistance response against the phytopathogenic bacterial infection, which needs to be clarified in the future [27]. For the other five flavonoids, kaemferol displayed moderate inhibitory activity only against the *S. aureus* among all the nine tested bacteria, and its MIC value was determined as 0.6 mg/mL. The isoorientin was more active towards *B. subtilis*, with a MIC value of 0.6 mg/mL. However, in this test, rutin, vitexin, and isovitexin exhibited weak or no obvious inhibitory activity against all nine representative bacteria, at the maximum concentration of 0.6 mg/mL, and their MIC values should be higher than 0.6 mg/mL. These results indicated that quercetin could be the main antibacterial constituent of the crude methanol extract (ME) of Tartary buckwheat sprout cultures, and the ME would be used as a potential antibacterial agent.

#### 2.2.2. Antifungal Activity of the Buckwheat Sprout Extract and Its Main Flavonoids

The spore germination assay results of the methanol extract (ME) of Tartary buckwheat sprouts, along with six main flavonoids standards against two representative pathogenic fungi, are presented in Figure 2. In general, the ME of Tartary buckwheat sprouts and four flavonoids (quercetin, rutin, isoorientin, and kaemferol) displayed notable antifungal activity, and the inhibitory efficiency was concentration dependent. As shown in Figure 2A, quercetin was the most active flavonoid for inhibiting the spore germination of *F. oxysporum* f. sp. *vasinfectum*, and its highest inhibitory rate was determined as much as 78.4%. Subsequently, the kaemferol, isoorientin, and rutin displayed moderate antifungal activity toward the spore germination of *F. oxysporum* f. sp. *vasinfectum*, and their inhibitory rate ranged from 26.1% to 49.8%. Moreover, the ME showed relatively weak inhibitory activity against the tested fungus, and its inhibitory rate was less than 10.0% in this investigation.

The antifungal activity of ME and four main flavonoids against *F. oxysporum* f. sp. *cucumerinum* had a similar pattern as their inhibitory efficacy toward the *F. oxysporum* f. sp. *vasinfectum*. As presented in Figure 2B, quercetin still exhibited the strongest inhibitory activity against the germination of *F. oxysporum* f. sp. *cucumerinum*, and its highest inhibitory rate of 81.2% was achieved within the test concentration of 0.2 mg/mL, followed by isoorientin and rutin, which showed moderate inhibitory activity against the *F. oxysporum* f. sp. *cucumerinum*, with the inhibitory rate of 47.2% and 39.7%, respectively. However, kaemferol and ME exhibited lower inhibitory activity against *F. oxysporum* f. sp. *cucumerinum*, and their inhibitory rate was less than 30%. Furthermore, vitexin and isovitexin did not show any inhibitory activity against the two pathogenic fungi, at the treatment concentration of 0.2 mg/mL in our test, therefore, they might not contribute to the antifungal activity of the crude methanol extract of Tartary buckwheat sprout cultures.

Based on the previous results, the median effective inhibitory concentration (IC_50_) values of the three potential active flavonoids (quercetin, isoorientin, and rutin) against the two pathogenic fungi were further evaluated. As listed in Table 3, quercetin exhibited strong activity for inhibiting the spore germination of *F. oxysporum* f. sp. *vasinfectum*, and *F. oxysporum* f. sp. *cucumerinum*, and its IC_50_ value was determined as 42.36 µg/mL and 32.85 µg/mL, respectively. In addition, rutin displayed moderate inhibitory activity toward the two tested fungi, and its IC_50_ values were calculated as 357.8 µg/mL and 257.32 µg/mL, respectively. Regarding the isoorientin concentration, it showed notable inhibitory ability against the test fungus of *F. oxysporum* f. sp. *cucumerinum*, with an IC_50_ value of 146.12 µg/mL. However, the inhibitory efficacy of isoorientin toward the *F. oxysporum* f. sp. *vasinfectum* was relatively weak, as its IC_50_ value should be higher than 400 µg/mL. Moreover, in this investigation, the *F. oxysporum* f. sp. *cucumerinum* was found to be more sensitive than *F. oxysporum* f. sp. *vasinfectum* to the ME of Tartary buckwheat sprouts and the main flavonoids.

Quercetin is a flavonoid well known for its antimicrobial potencies, and it has been successfully obtained from many medicinal plants. Several studies have demonstrated the notable antimicrobial activity of quercetin. Lu et al. reported that the quercetin exhibited strong activity against the growth of phytopathogenic fungus *Helminthosporium sativum*, with a MIC value of 50 µg/mL. However, it was inactive toward the human pathogenic fungi *Candida albicans* and *Trichophyton rubrum* [28]. The crude methanol extracts of *Thymbra spicata* var. *spicata* and *Origanum minutiflorum* exhibited good antibacterial activity against the *Mycobacterium tuberculosis*, with MIC values of 196.0 µg/mL and 392.0 µg/mL, respectively, and quercetin was confirmed to be the main active compound of these extracts [29]. Boligon et al. reported the promising anti-*Mycobacterium smegmatis* activity of the ethyl acetate extract of *Scutia buxifolia* leaves, with a MIC value of 312.5 µg/mL, and quercetin and quercitrin were found to be the main active components of the crude extract [30]. In this study, the antimicrobial activity assays revealed that quercetin displayed excellent antimicrobial activity toward most of the tested bacteria and fungi, and it could mostly be the key component contributing to the antimicrobial activity of the extract of Tartary buckwheat sprouts. 

### 2.3. DPPH Radical Scavenging Activity

The DPPH radical scavenging ability of the crude methanol extract (ME) of Tartary buckwheat sprouts and six main flavonoids is shown in Figure 3A. Generally, the crude ME of Tartary buckwheat sprouts and six flavonoids, used as reference standards, except for vitexin, displayed notable antioxidant activity, and the radical clearance effect was concentration dependent. Of them, quercetin, rutin, and kaemferol exhibited relatively higher radical scavenging ability than other tested samples. At the test concentration of 100 µg/mL, the DPPH radical scavenging rates of these three flavonoids were all over 85%.

According to the results mentioned above, the IC_50_ values of the crude methanol extract of Tartary buckwheat sprouts and four potential active flavonoids (quercetin, kaemferol, rutin, and isoorientin) for clearing the DPPH radical were investigated further. As displayed in Figure 3B, quercetin exhibited the strongest activity for the DPPH radical scavenging test, followed by kaemferol and rutin. Correspondingly, their IC_50_ values were calculated as 5.60 µg/mL, 16.23 µg/mL, and 27.95 µg/mL, respectively, which were all lower than the positive control (trolox) of 40.21 µg/mL. Quercetin, rutin, and kaemferol could remove the DPPH radical effectively, having important roles in the antioxidant systems of Tartary buckwheat sprouts. The crude methanol extract of buckwheat sprouts and isoorientin displayed moderate activity in the DPPH radical scavenging assay, and their IC_50_ values were determined as 162.24 µg/mL and 181.71 µg/mL, respectively. Both vitexin and isovitexin showed relatively weak activity in the DPPH radical scavenging test, with clearance rates less than 20%, and their IC_50_ values were not detected.

### 2.4. Structure–Activity Relationship (SAR) of the Flavonoids

In the antimicrobial activity investigations, quercetin was screened to be the most active among the six major flavonoids of Tartary buckwheat sprouts. It has been reported that, in the molecular structure of flavonoids (Figure 1B), 3′ and 4′ ortho-dihydroxylation on the B ring could strongly affect its bioactivities, and the hydroxyl substitution at the third position on the C ring might reduce the inhibitory activity [31,32]. Vitexin, isovitexin, and kaemferol exhibited relatively lower effects in the antimicrobial activity assay, which is probably due to the paucity of 3′and 4′ dihydroxyl groups on their structures. Additionally, the antioxidant activity of quercetin, kaemferol, rutin, and isoorientin was stronger than that of the vitexin or isovitexin in the DPPH radical scavenging test. It also argued that the hydroxyl at the third position on the C ring could be a crucial element for antioxidant ability. Flavonoids (i.e., quercetin and kaemferol) that had hydroxyl at the third position on the C ring displayed efficacious radical scavenging activity. Nevertheless, it was certainly confirmed that quercetin could be the key bioactive flavonoid that contributed to the antimicrobial and antioxidant activities of the crude methanol extract of Tartary buckwheat sprout cultures. These results could help enrich the pharmacological features and reveal the functions of flavonoids in Tartary buckwheat.

## 3. Experimental 

### 3.1. Preparation of Tartary Buckwheat Sprout Cultures

The healthy Tartary buckwheat seeds (cultivar chuanqiao-01) were selected and washed with running water for 1 min, then sterilized in 0.5% sodium hypochlorite solution for 5 min. After washing with distilled water three times, the prepared seeds were placed onto the germination boxes with three-layer filter paper. Then, Tartary buckwheat sprout cultures were cultivated in illumination incubators (LEDs, 7500 lx) at 25 ± 1 °C and 70% relative humidity, and harvested on day 10. 

### 3.2. Preparation of the Crude Extracts of Tartary Buckwheat Sprouts

A total of 294.1 g fresh Tartary buckwheat sprout cultures were harvested on day 10. After drying at 40–45 °C in a drying oven (DGG-9246A, Qixin, Shanghai, China) to a constant dry weight, about 37.3 g of buckwheat sprouts were obtained. Then, the sprout cultures were ground into powder and extracted with 70% methanol–water solution under sonication for 30 min, and the ratio of material to liquid was 1:50 (*m*/*v*). The supernatant was collected by filtration, and the extraction procedure was repeated three times. All of the supernatant content was concentrated under vacuum at 30–35 °C by a rotary evaporator and afforded the crude methanol extract (ME, 13.1 g, shown in Figure 1A) of Tartary buckwheat sprouts, and its extraction yield (*g*/*g*) was determined as 35.1%. The crude extract was stored at 4 °C and kept in the dark prior to use. 

### 3.3. Determination of Total Flavonoid Contents of Tartary Buckwheat Sprouts

The total flavonoid content of the sprouts extract was determined according to the methods described in our previous study [33]. Generally, the total flavonoid content was measured by the aluminum trichloride colorimetry method and expressed as milligram of rutin equivalent (RE) per gram of the extract. 

### 3.4. Analysis and Quantification of Major Flavonoids from Tartary Buckwheat Sprouts

In order to figure out the main flavonoids in Tartary buckwheat sprouts, the crude extract (ME) of buckwheat sprouts was subjected to high-performance liquid chromatography (HPLC) analysis. For this procedure, 2.0 mg of ME or the selected standard pure compound was diluted in 1 mL of methanol and diluted to different concentrations. For determination of the major flavonoids content of Tartary buckwheat sprout cultures, the extraction was carried out by mixing buckwheat sprout powder (0.1 g) with a methanol-water (25 mL, 70%, *v*/*v*) solution in a conical flask under ultrasonic processing for 30 min. After filtration, the filtrates were transferred into a 25 mL volumetric flask, and the volume was adjusted to 25 mL with 70% methanol solution. A 0.45 μm filtrate membrane was applied to remove the particles and microbes. Sample analysis was performed on an LC-10ATvP system, equipped with an SPD-M10AvP diode-array detector (Shimadzu, Kyoto, Japan), and a C_18_ column (4.6 mm × 250 mm, 5 μm, Phenomenex, Torrance, CA, USA). The mobile phase was A (water with 1% acetic acid) and B (methanol with 1% acetic acid). The elution program was as follows: 0–2 min 15% B; 2.01–8 min 15–35% B; 8.01–12 min 35% B; 12.01–40 min 35–90% B; 40.01–45 min 15% B; 45.01–50 min 100% B. The flow rate was 0.8 mL/min at 30 °C, and the injection volume was 20 μL. The wavelength of detection was 365 nm. Six pure flavonoids—isoorientin, vitexin, isovitexin, rutin, quercetin, and kaemferol—were used as reference standards, which were purchased from the Sichuan Vicky Biotechnology Co., Ltd. Identification of the flavonoids was achieved by comparing the retention time of samples to those of the standards, and referring to related references [20]. Quantification of the flavonoids was obtained by using an external standard method [34]. 

### 3.5. Antimicrobial Activity Test

#### 3.5.1. Antibacterial Activity Evaluation of Buckwheat Sprout Extract and Main Flavonoids

The antibacterial activity of the crude methanol extract (ME) of Tartary buckwheat sprouts and six major flavonoids (isoorientin, vitexin, isovitexin, rutin, quercetin, and kaemferol) were conducted according to a modified colorimetric broth microdilution method [35]. A total of nine tested bacteria, comprising five Gram-negative and four Gram-positive strains, were selected—namely, *Escherichia coli* ATCC 29425, *Salmonella typhimurium* CMCC 50115, *Agrobacterium tumefaciens* ATCC 11158, *Pseudomonas lachrymans* ATCC 11921, *Xanthomonas vesicatoria* ATCC 11633, *Bacillus subtilis* ATCC 11562, *Staphylococcus albus* CICC 10897, *S. epidermidis* ATCC 12228, and *S. aureus* ATCC 6538. Generally, the tested bacteria were grown in Luria–Bertani (LB) media, which contained yeast extract 5.0 g/L, peptone 10.0 g/L, NaCl 5.0 g/L, and pH 7.2. The sprout methanol extract (ME) and six main flavonoids were dissolved in 60% methanol and diluted to different concentrations ranging from 3200 to 400 μg/mL. Then, 10 μL of test sample solutions and 90 μL of prepared bacterial suspension (1 × 10^6^ CFU/mL) were added into each well of the 96-well microplate and mixed by a plate shaker. The plant pathogenic bacteria were incubated at 25 °C, and the clinic strains at 37 °C, respectively. Streptomycin sulfate was used as the positive control, and 60% methanol was as the negative control. The minimum inhibitory concentration (MIC) value and minimum bactericidal concentration (MBC) value were also determined. All treatments were repeated three times.

#### 3.5.2. Antifungal Activity Assessment of Buckwheat Sprout Extract and Main Flavonoids

Two representative *Fusarium* samples (*Fusarium oxysporum* f. sp. *vasinfectum* and *F. oxysporum* f. sp. *cucumerinum*), kindly provided by Professor Ligang Zhou, College of Plant Protection, China Agriculture University, were selected as the tested fungi. A modified spore germination assay was used to determine the antifungal activity of ME and six flavonoids, used as reference standards, according to our previous study [36]. Briefly, the test fungi were grown in potato dextrose (PD) liquid media (potato 200.0 g/L, dextrose 20.0 g/L) and cultivated on a rotary shaker (25 °C, 120 rpm) for 4–7 days in the dark. The fungal cultures were filtrated with an aseptic cloth, and the filtrate was centrifuged to obtain the spores. Sterilized water was used to resuspend the precipitates and dilute them to 1 × 10^6^ spores/mL. Then, 5 μL sample solutions and 45 μL fungal spore suspension were added onto concave glass slides and incubated in a moist chamber at 25 °C for 7 h. Series solutions with different concentrations of ME and flavonoids were prepared in 30% methanol for the test. Carbendazim was used as the positive control, 30% methanol was used as the negative control, and the sterilized water was the blank. All tests were conducted in triplicate. After incubation, slides were examined under a microscope to examine the status of spores. The inhibitory rate of spore germination was determined, and the median effective inhibitory concentration (IC_50_) was calculated according to the method of Sakuma [37]. 

### 3.6. DPPH Radical Scavenging Assay

The DPPH (1,1-diphenyl-2-picrylhydrazyl) radical scavenging test was conducted in a 96-well-microplate according to the method of Wan-Nadilah [38]. In brief, the crude methanol extract (ME) of Tartary buckwheat sprouts and six main flavonoids (isoorientin, vitexin, isovitexin, rutin, quercetin, and kaemferol), used as reference standards, was dissolved in methanol and diluted to series concentrations of 500 µg/mL, 250 µg/mL, and 125 µg/mL, respectively. Then, 20 µL of sample solutions and 80 µL DPPH solutions of 0.2 mg/mL were added into each well of the microplate. The plates were incubated at 37 °C for 30 min and measured at the absorbance of 515 nm by a microplate spectrophotometer. The inhibition rate of free radical was calculated, and the IC_50_ value was also determined. The Butylated hydroxytoluene (BHT) was used as the positive control, and methanol was used as the negative control. All tests were run in triplicate. 

### 3.7. Statistical Analysis

All tests were performed in triplicate, and the results were expressed by their mean values and standard deviation (SD). The experimental data were submitted to analysis of their variance to determine significant differences by PROC ANOVA of SAS version 9.2 (SAS Institute Inc., Cary, NC, USA). The term significant difference was based on *p* ≤ 0.05.

## 4. Conclusions

The crude methanol extract of Tartary buckwheat sprout cultures was abundant in flavonoids. Six main flavonoids (i.e., isoorientin, vitexin, isovitexin, rutin, quercetin, and kaemferol) were confirmed and quantified from Tartary buckwheat sprouts by the HPLC analysis method. Rutin and quercetin appeared at relatively higher contents, while isoorientin and kaemferol were present in trace amounts in Tartary buckwheat sprout cultures. To the best of our knowledge, this is the first report on the antimicrobial activity of the crude methanol extract of Tartary buckwheat sprout cultures. The antimicrobial activity assays revealed that quercetin displayed excellent antimicrobial activity toward most of the tested bacteria and fungi, and it should be the main active component of the methanol extract of buckwheat sprouts for contributing to the antimicrobial activity. Quercetin, rutin, and kaemferol could remove the DPPH radical effectively, having important roles for the sprout antioxidant systems. These results could help enrich the pharmacological characteristics and understand the potential functions of flavonoids in Tartary buckwheat. Furthermore, it is more beneficial for making full use of Tartary buckwheat sprouts, as well as the sprouts extract or its main flavonoids, in the fields of nutrition, pharmacology, agriculture, and food science.

## Figures and Tables

**Figure 1 molecules-27-00374-f001:**
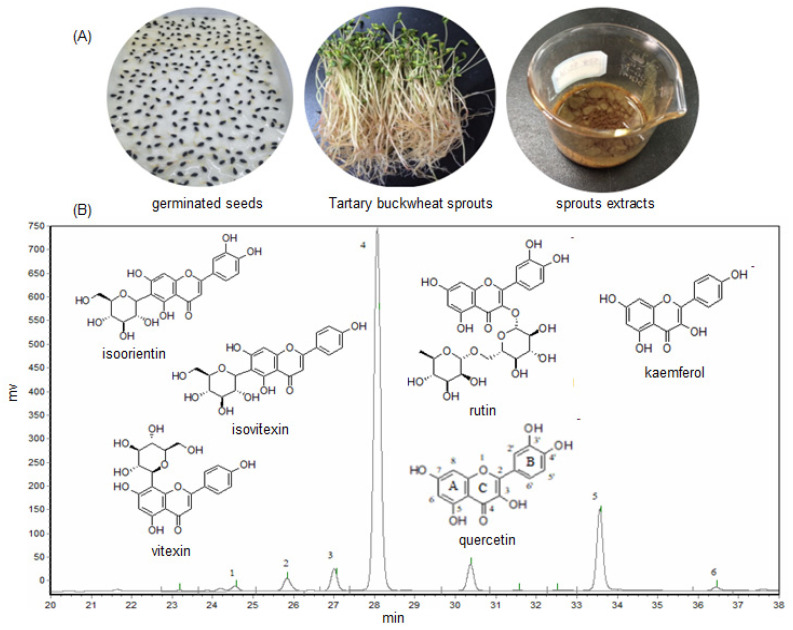
The germinated seeds, sprouts, and methanol extract (**A**), and the typical HPLC chromatogram of methanol extract (**B**) of Tartary buckwheat sprout cultures. Here, 1—isoorientin, 2—vitexin, 3—isovitexin, 4—rutin, 5—quercetin, 6—kaemferol.

**Figure 2 molecules-27-00374-f002:**
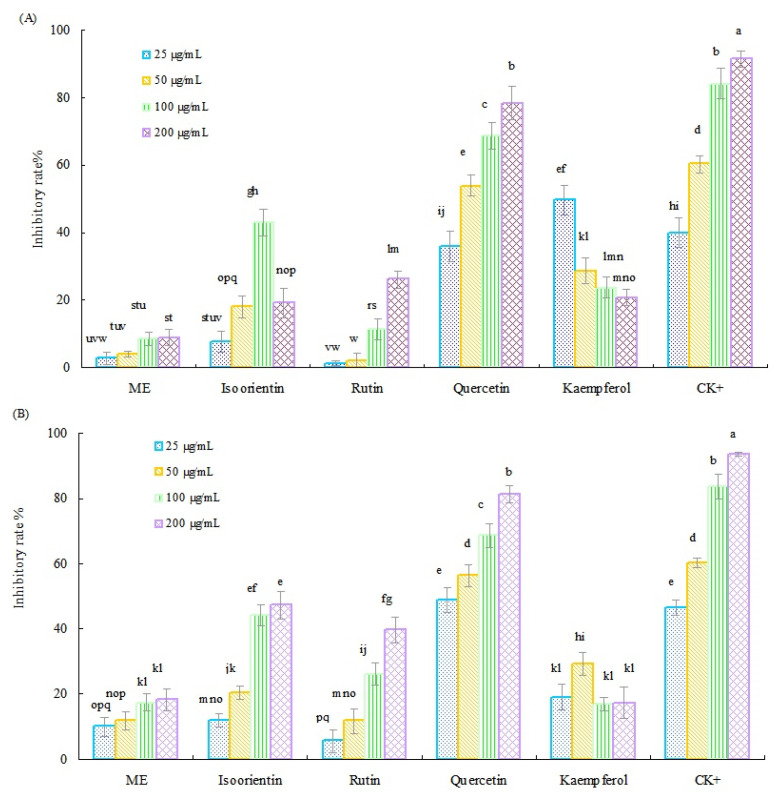
Inhibitory activity of ME of Tartary buckwheat sprouts and four flavonoids against the spore germination of *F. oxysporum* f. sp. *vasinfectum* (**A**) and *F. oxysporum* f. sp. *cucumerinum* (**B**). CK^+^ was carbendazim. Different letters (i.e., a–q) in each column indicated significant differences among the treatment at *p* = 0.05 level.

**Figure 3 molecules-27-00374-f003:**
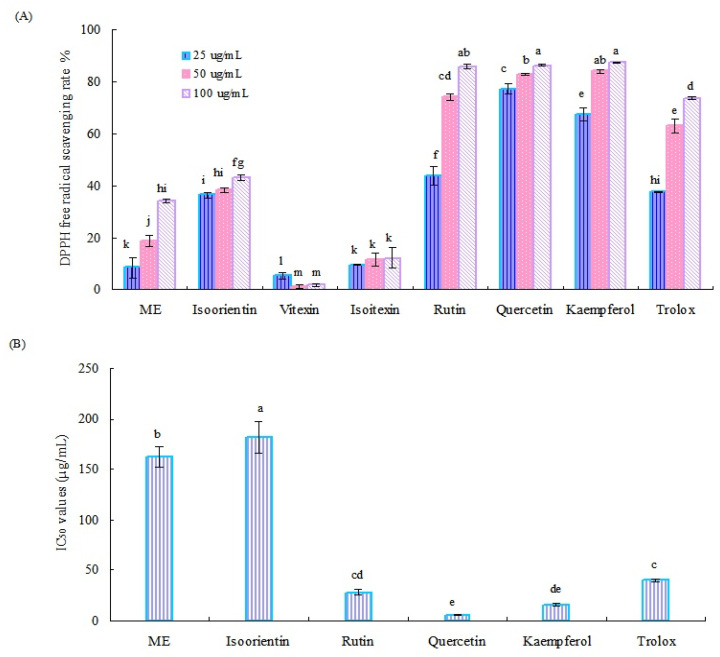
DPPH radical scavenging rate (**A**) and IC_50_ values (**B**) of the methanol extract of Tartary buckwheat sprouts and six main flavonoids. Different letters (i.e., a–m) in each column indicated significant differences among the treatment samples, at *p* = 0.05 level.

**Table 1 molecules-27-00374-t001:** The flavonoid content of the crude extract and sprout cultures of Tartary buckwheat.

Compound	Retention Time (min)	Regress Equation (Y = aX + b)	Determination Coefficient (R^2^)	Content in the Extract (mg/g dw)	Content in the Sprouts (mg/g dw)
Isoorientin	24.555	Y = 65,468X + 26,976	0.9998	0.92 ± 0.04	0.32 ± 0.01
Vitexin	25.837	Y = 49,928X + 99,712	0.9990	2.84 ± 0.23	0.98 ± 0.08
Isovitexin	27.002	Y = 45,270X – 28,163	0.9996	6.28 ± 0.16	2.20 ± 0.06
Rutin	28.068	Y = 49,996X + 107,063	0.9992	89.81 ± 0.41	31.50 ± 0.14
Quercetin	33.568	Y = 43,830X – 60,890	0.9996	23.34 ± 2.54	8.17 ± 0.89
Kaemferol	36.438	Y = 68,692X – 16,938	0.9998	0.86 ± 0.01	0.30 ± 0.00

**Table 2 molecules-27-00374-t002:** The MIC values of crude methanol extract of Tartary buckwheat sprouts and six major flavonoids against test bacteria.

Test bacterium	ME	Isoorientin	Vitexin	Isovitexin	Rutin	Quercetin	Kaemferol	CK^+^
*A. tumefaciens*	>3.2	>0.60	>0.60	>0.60	>0.60	0.20	>0.60	0.04
*E. coli*	>3.2	>0.60	>0.60	>0.60	>0.60	>0.60	>0.60	0.04
*P. lachrymans*	0.8	>0.60	>0.60	>0.60	>0.60	0.40	>0.60	0.04
*X. vesicatoria*	>3.2	>0.60	>0.60	>0.60	>0.60	0.20	>0.60	0.04
*S. typhimurium*	1.6	>0.60	>0.60	>0.60	>0.60	0.60	>0.60	0.04
*B. subtilis*	1.6	0.60	>0.60	>0.60	>0.60	0.40	>0.60	0.04
*S. albus*	3.2	>0.60	>0.60	>0.60	>0.60	0.40	>0.60	0.04
*S. aureus*	3.2	>0.60	>0.60	>0.60	>0.60	0.60	0.60	0.10
*S. epidermidis*	>3.2	>0.60	>0.60	>0.60	>0.60	>0.60	>0.60	0.04

CK^+^, the Streptomycin sulfate; MIC, the minimum inhibitory concentration (mg/mL).

**Table 3 molecules-27-00374-t003:** The IC_50_ values of quercetin, rutin, and isoorientin against the spore germination of two pathogenic fungi.

Samples	Test Fungi	Regress Equation	Determination Coefficient (R^2^)	IC_50_ (µg/mL)
Quercetin	*F. oxysporum* f. sp. *vasinfectum*	Y = 2.3272X + 1.2137	0.9785	42.4 ± 2.6
	*F. oxysporum* f. sp. *cucumerinum*	Y = 1.7053X + 2.4138	0.9421	32.9 ± 1.4
Rutin	*F. oxysporum* f. sp. *vasinfectum*	Y = 2.3537X − 1.0019	0.9948	357.8 ± 15.5
	*F. oxysporum* f. sp. *cucumerinum*	Y = 1.6574X + 1.0029	0.9767	257.3 ± 23.42
Isoorientin	*F. oxysporum* f. sp. *vasinfectum*	nd	nd	>400
	*F. oxysporum* f. sp. *cucumerinum*	Y = 0.3746X + 3.4929	0.9530	146.1 ± 9.8
CK^+^	*F. oxysporum* f. sp. *vasinfectum*	Y = 1.4235X + 2.9076	0.9896	29.5 ± 3.1
	*F. oxysporum* f. sp. *cucumerinum*	Y = 1.2563X + 3.1901	0.9849	27.6 ± 2.5

CK^+^, carbendazim; nd, not detected.

## Data Availability

The data presented in this study are available on request from the corresponding author.

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
