# Peer review of "Chemical Profile, Antimicrobial and Antioxidant Activity Assessment of the Crude Extract and Its Main Flavonoids from Tartary Buckwheat Sprouts"

_molecules, 2022, doi:10.3390/molecules27020374_

Round 1

Reviewer 1 Report

The paper entitled: „Chemical Profile, Antimicrobial and Antioxidant  Activity Assessment of the Crude Extract and Its Main Flavonoids from Tartary Buckwheat Sprouts” by Lingyun Zhong, Yuji Lin, Can Wang, Bei Niu , Ying Xu , Gang Zhao and Jianglin Zhao is intereting and valuable.  However, some errors and ambiguities have been noted in this work. I present my comments and suggestions below:

Abstract

The first sentence of the abstract

The purpose of this study was to investigate the major flavonoids compositions and 15 bioactivities of the Tartary buckwheat sprouts.

this sentence should read rather like this

The purpose of the study was to investigate the content of flavonoids and the bioactivity of tartar buckwheat sprouts.

The highest rutin content in the ME and sprouts was 89.81 mg/g and 31.50 mg/g, respectively.

What does rutin content in methanol extract and sprouts mean? So two extractions were made? How can the content of a reference substance in sprouts be measured? This needs to be clarified and corrected.

These results indicated that the methanol extract and its main active 35 flavonoids of Tartary buckwheat sprouts could be used as a potential antimicrobial or antioxidant 36 agent in the future.

I think that we should focus on the biological activity of the extract, because, as we know from the available literature, the activity of flavonoids has already been confirmed many times. The fragment „…and its main active 35 flavonoids…” should be removed.

  1. Results and Discussion

 2.1. Chemical Profile of the Crude Extract of Tartary Buckwheat Sprouts

„A total of 294.1 g fresh Tartary buckwheat sprout cultures were harvested on day 10. After 82 drying in an oven to a constant dry weight, about 37.3 g of buckwheat sprouts were obtained. 83 Through extraction, filtration, and concentration, about 13.1 g of the crude methanol extract (shown 84 in Figure 1A) of Tartary buckwheat sprouts was attained, and its extraction yield (g/g) was 85 determined as 35.1%”

In my opinion, this text should be included in the experimental part

2.1. Chemical Profile of the Crude Extract of Tartary Buckwheat Sprouts

In line with the title of the paragraph, the chemical profiles of the tested extracts are discussed here, the remaining information (total content of polyphenols and flavonoids) should be placed in the paragraph after the antioxidant activity, because they are closely related to each other

The values for the retention times of the reference substances should be added in the text or in the table.

Were patterns identified by retention times only? The spectra of these substances should also be compared.

2.3. DPPH Radical Scavenging Activity

data on the total phenolics and total flavonoids should be included in the section on antioxidant activity, i.e. here

The data on the total content of polyphenols was not found in the text, these data should be completed and added in the form of a table (similar to Table 1)

3.6. DPPH Radical Scavenging Assay

… the crude methanol extract (ME) or the main six flavonoids of Tartary buckwheat sprouts were ….

First: Wrong wording, have both the extract and the reference substances been tested? Instead of "or" write "i".

Second: You shouldn't write the six flavonoids from the sprouts because it looks like these substances were isolated from the sprouts…. It should simply be written that apart from the extract, these six substances were also analyzed.

Conclusions

„Six main flavonoids (i.e., isoorientin, vitexin, isovitexin, rutin, quercetin, and kaemferol) were confirmed and quantified from the Tartary buckwheat sprouts by the HPLC analysis method.”

In my opinion, there is no quantification of the standard substances in the extract in this work.

Author Response

Reviewer: 1

Comments to the author

The paper entitled: Chemical Profile, Antimicrobial and Antioxidant  Activity Assessment of the Crude Extract and Its Main Flavonoids from Tartary Buckwheat Sprouts” by Lingyun Zhong, Yuji Lin, Can Wang, Bei Niu , Ying Xu , Gang Zhao and Jianglin Zhao is intereting and valuable.  However, some errors and ambiguities have been noted in this work. I present my comments and suggestions below:

  1. Abstract

(1) The first sentence of the abstract,

The purpose of this study was to investigate the major flavonoids compositions and 15 bioactivities of the Tartary buckwheat sprouts.

this sentence should read rather like this,

The purpose of the study was to investigate the content of flavonoids and the bioactivity of tartar buckwheat sprouts.

Response: Thank you very much. We have rephrased this sentence to make it more accurate according to your valuable suggestion.

(2) The highest rutin content in the ME and sprouts was 89.81 mg/g and 31.50 mg/g, respectively.

What does rutin content in methanol extract and sprouts mean? So two extractions were made? How can the content of a reference substance in sprouts be measured? This needs to be clarified and corrected.

Response: Thanks. In fact, there are two different extraction samples, which have been successfully prepared for determination the content of rutin by HPLC analysis method in this investigation.

For preparation of the crude methanol extract (ME, sample 1) of Tartary buckwheat sprouts, the harvested sprouts were dried at at 40-45°C in a drying oven, and were ground into powder. Then, the sprouts powder (37.3 g) was extracted with 70% methanol- water solution (about 2000 mL) under sonication for 30 min (the ratio of material to liquid was 1:50, m/v), and the supernatant was collected by filtration. Next, all the supernatant was concentrated under vacuum at 30-35 °C by a rotary evaporator to dryness, and afforded the methanol extract (ME) of tartary buckwheat sprouts. The crude extract was stored at 4 °C and kept in the dark prior to analyze the flavonoids content.

For preparation of the methanol extraction (sample 2) of Tartary buckwheat sprouts, the extraction was carried out by mixing buckwheat sprout powder (0.1 g) with a methanol-water (25 mL, 70%, v/v) solution in a conical flask under ultrasonic processing for 30 min. After filtration, the filtrates were transferred into a 25 mL volumetric flask and the volume adjusted to 25 mL with 70% methanol solution. Then, the extraction (without concentration and evaporation) of buckwheat sprouts was directly subjected to HPLC analysis of the flavonoids content by the HPLC method mentioned in our previous reports.

As far as we know, the HPLC method has been widely used for analyzing the content of known components in many fields. In this research, the HPLC method for analysis of the six major flavonoids (rutin, quecetin, isovitexin, vitexin, isoorientin, kaemferol) of Tartary buckwheat had been successfully established in our previous researches, and it has been successfully and widely used for analyzing the flavonoids in various Tartary buckwheat or common buckwheat samples in our laboratory for a long time (following references). Therefore, it should be a scientific and reliable method for analysis the major flavonoids (including rutin) of Tartary buckwheat sprouts in this study.

(1) Wei L, Chen G, Yang C, Zhang Y, Yao Z, Zhao G, Peng L. Simultaneous  quantitative analysis of six components by quantitative analysis of muti-components by single marker method in buckwheat. Food Science and Technology, 2019, 44(3): 303-308.

(2) Li W. Preliminary study on the storage and preservation condition of Tartary buckwheat sprout. Chengdu University Dissertation, 2018.

(3) Zhong L, Niu B, Tang L, Chen F, Zhao G, Zhao J. Effects of polysaccharide elicitor from endophytic Fusarium oxysporum Fat9 on the growth , flavonoid accumulation and antioxidant property of Fagopyrum tataricum sprout cultures. Molecules, 2016, 21: 1590.

 (3) These results indicated that the methanol extract and its main active 35 flavonoids of Tartary buckwheat sprouts could be used as a potential antimicrobial or antioxidant 36 agent in the future.

I think that we should focus on the biological activity of the extract, because, as we know from the available literature, the activity of flavonoids has already been confirmed many times. The fragment „…and its main active 35 flavonoids…” should be removed.

Response: Thanks very much. We have rephrased this sentence to make it more accurate according to your helpful suggestion.

  1. Results and Discussion

(1) 2.1. Chemical Profile of the Crude Extract of Tartary Buckwheat Sprouts

„A total of 294.1 g fresh Tartary buckwheat sprout cultures were harvested on day 10. After 82 drying in an oven to a constant dry weight, about 37.3 g of buckwheat sprouts were obtained. 83 Through extraction, filtration, and concentration, about 13.1 g of the crude methanol extract (shown 84 in Figure 1A) of Tartary buckwheat sprouts was attained, and its extraction yield (g/g) was 85 determined as 35.1%”

In my opinion, this text should be included in the experimental part

Response: Thanks very much. We have rephrased these sentences, and they have been moved to the experimental part.

(2) 2.1. Chemical Profile of the Crude Extract of Tartary Buckwheat Sprouts

In line with the title of the paragraph, the chemical profiles of the tested extracts are discussed here, the remaining information (total content of polyphenols and flavonoids) should be placed in the paragraph after the antioxidant activity, because they are closely related to each other

The values for the retention times of the reference substances should be added in the text or in the table.

Were patterns identified by retention times only? The spectra of these substances should also be compared.

Response: Thanks. We have added the retention times of the six flavonoids, isoorientin (24.555 min), vitexin (25.837 min), isovitexin (27.002 min), rutin (28.068 min), quercetin (33.568 min), and kaemferol (36.438 min) in the Table 1 to make it more informative to the readers according to your valuable suggestions.

As we know, the MS, IR, and 1H-NMR, 13C-NMR, 1H-1H COSY, HMBC, HMQC, NOESY, and other spectral or physicochemical evidence are widely employed to elucidate and identify an objective compound (especial for an unknown compound). In our laboratory, the HPLC method, as well as the LC-MS method for determination and analysis the major flavonoids or phenolics (known constituents) of various buckwheat samples (i.e., buckwheat seeds, sprouts, buckwheat tea, buckwheat noodles, and so on) was successfully established, and it had been proved to be a scientific and reliable method (following references). Therefore, the HPLC method was applied for analysis of the main flavonoids content of the crude methanol extract of Tartary buckwheat sprouts in this research. In the future research program, some physicochemical evidence and spectral method would be applied to elucidate and clarify the bioactive components, especial for the unknown compound of buckwheat. Thanks again.

(1) Wei L, Chen G, Yang C, Zhang Y, Yao Z, Zhao G, Peng L. Simultaneous  quantitative analysis of six components by quantitative analysis of muti-components by single marker method in buckwheat. Food Science and Technology, 2019, 44(3): 303-308.

(2) Li W. Preliminary study on the storage and preservation condition of Tartary buckwheat sprout. Chengdu University Dissertation, 2018.

(3) Zhong L, Niu B, Tang L, Chen F, Zhao G, Zhao J. Effects of polysaccharide elicitor from endophytic Fusarium oxysporum Fat9 on the growth , flavonoid accumulation and antioxidant property of Fagopyrum tataricum sprout cultures. Molecules, 2016, 21: 1590.

(4) Zhong L. Bioactivity evaluation of flavonoid constituents and the effects of endophytic fungi on the growth and flavonoids accumulation of tartary buckwheat sprout cultures. Sichuan University Dissertation, 2016.

(5) Zhao J, Zou L, Zhong L, Peng L, Ying P, Tan M, Zhao G. Effects of polysaccharide elicitors from endophytic Bionectria pityrodes Fat6 on the growth and flavonoid production in tartary buckwheat sprout cultures. Cereal Research Communication, 2015, 43, 661-671.

(3) 2.3. DPPH Radical Scavenging Activity

data on the total phenolics and total flavonoids should be included in the section on antioxidant activity, i.e. here

The data on the total content of polyphenols was not found in the text, these data should be completed and added in the form of a table (similar to Table 1)

Response: Thanks very much. In this study, both the total phenolics and total flavonoids content in the crude methanol extract (ME) of Tartary buckwheat sprouts were preliminary measured firstly. Although, the main purpose of this investigation was to clarify the major flavonoids compositions of the ME and sprout cultures of Tartary buckwheat, and evaluate their bioactivities (antimicrobial and antioxidant activities). Then, the phenolic components were not been detected and analyzed in detail. We will carry out a research program on clarification the chemical composition of major phenolics, as well as their bioactivities in the future. Therefore, we have deleted the experimental method and result section on the total phenolics of the ME of Tartary buckwheat sprout crude extract, and revised these sentences to make it more accurate according to your suggestions.

(4) 3.6. DPPH Radical Scavenging Assay

… the crude methanol extract (ME) or the main six flavonoids of Tartary buckwheat sprouts were ….

First: Wrong wording, have both the extract and the reference substances been tested? Instead of "or" write "i".

Second: You shouldn't write the six flavonoids from the sprouts because it looks like these substances were isolated from the sprouts…. It should simply be written that apart from the extract, these six substances were also analyzed.

Response: Thanks very much. In fact, the DPPH radical scavenging ability of both the crude methanol extract (ME) of Tartary buckwheat sprouts and six flavonoid standards were tested in this research. We have rephrased this sentence (the crude methanol extract (ME) of Tartary buckwheat sprouts, and six main flavonoid standards (isoorientin, vitexin, isovitexin, rutin, quercetin, and kaemferol)…) to make it more accurate according to your suggestions.

  1. Conclusions

„Six main flavonoids (i.e., isoorientin, vitexin, isovitexin, rutin, quercetin, and kaemferol) were confirmed and quantified from the Tartary buckwheat sprouts by the HPLC analysis method.”

In my opinion, there is no quantification of the standard substances in the extract in this work.

Response: Thanks. In fact, the six main flavonoids (i.e., isoorientin, vitexin, isovitexin, rutin, quercetin, and kaemferol) were successfully confirmed and quantified from the Tartary buckwheat sprouts by the HPLC analysis method, and their content in the crude methanol extract (ME) and in the sprout culture was displayed in Table 1, respectively.

Reviewer 2 Report

Article
Chemical Profile, Antimicrobial and Antioxidant Activity Assessment of the Crude Extract and Its Main Flavonoids from Tartary Buckwheat Sprouts

A brief summary
In this work content of 6 flavonoids in crude methanol extract of Tartary buckwheat sprouts and it’s bioactivities was evaluated. The research is quite well designed, performed, analysed and described although it needs some improvements. The paper is quite interesting, unfortunately the scope of the research and the techniques used are not sufficient for publication in a scientific journal.

Broad comments

1. In several places (i.e. line 16) the authors mention that buckwheat sprouts contain phenolic AND flavonoids. Why then were they not detected in extracts tested by HPLC? Why does the statement 'Tartary buckwheat sprout cultures was abundant in phenols and flavonoids' (line 331) appear in the conclusions if they were not detected and determined in the tests carried out?

2. Validation of the HPLC method is lacking. If the method described in publication no. 34 was used, this should be indicated. The validation elements included in Table 1. are insufficient and should in principle not be presented behind the results.

3. The description of the extraction is insufficient. Was 70 % methanol (line 254) used? How much was raw material and how much solvent? Were the extracts evaporated to dryness (line 256)?

4. Key words need improvement.

Specific comments

lines 50, 247 –  How was seed health tested?

lines 82, 253 – What type of dryer is it?

line 248 – sodium hypochlorite?

line 251 – Which lamps, which light intensity?

line 271 – Is it really about removing MICROBES from samples for chromatographic separation?

Author Response

Reviewer: 2

Comments to the Author

This Article
Chemical Profile, Antimicrobial and Antioxidant Activity Assessment of the Crude Extract and Its Main Flavonoids from Tartary Buckwheat Sprouts

A brief summary

In this work content of 6 flavonoids in crude methanol extract of Tartary buckwheat sprouts and it’s bioactivities was evaluated. The research is quite well designed, performed, analysed and described although it needs some improvements. The paper is quite interesting, unfortunately the scope of the research and the techniques used are not sufficient for publication in a scientific journal.

Broad comments

  1. In several places (i.e. line 16) the authors mention that buckwheat sprouts contain phenolic AND flavonoids. Why then were they not detected in extracts tested by HPLC? Why does the statement 'Tartary buckwheat sprout cultures was abundant in phenols and flavonoids' (line 331) appear in the conclusions if they were not detected and determined in the tests carried out?

Response: Thank you very much. Many researchers have reported that the Tartary buckwheat seeds or sprouts were abundant in phenols and flavonoids. In this research, both the total phenolics and total flavonoids content in the crude methanol extract (ME) of Tartary buckwheat sprouts were preliminary measured firstly. Although, the main purpose of this investigation was to clarify the major flavonoids compositions of the ME and sprout cultures of Tartary buckwheat, and evaluate their antimicrobial and antioxidant activities. Then, the phenolic components were not been detected and analyzed in detail. We will carry out a research program on clarification the chemical composition of major phenolics, as well as their bioactivities in the future. Therefore, we have deleted the experimental method and result section of total phenolics of the ME of Tartary buckwheat sprout crude extract, and revised these sentences to make it more accurate according to your suggestions.

  1. Validation of the HPLC method is lacking. If the method described in publication no. 34 was used, this should be indicated. The validation elements included in Table 1. are insufficient and should in principle not be presented behind the results.

Response: Thanks very much. As far as we know, the HPLC method has been widely used for analysis of the content of known components in many fields. To establish a reliable HPLC analysis method, a few research program (such as the retention time test, linear relation test, stability test, precision test, repeatability test, recovery test, and so on) should be successfully performed. In this research, the HPLC method for analysis of the major flavonoids of Tartary buckwheat had been successfully established in our previous researches (many validation tests had been performed, data not shown), and it has been successfully and widely used for analyzing the flavonoids in various Tartary buckwheat or common buckwheat samples (i.e., buckwheat seeds, sprouts, buckwheat tea, buckwheat noodles, and so on) in our laboratory for a long time (following references). Therefore, it should be a scientific and reliable method for analysis the major flavonoids of Tartary buckwheat sprouts in this study.

(1) Wei L, Chen G, Yang C, Zhang Y, Yao Z, Zhao G, Peng L. Simultaneous  quantitative analysis of six components by quantitative analysis of muti-components by single marker method in buckwheat. Food Science and Technology, 2019, 44(3): 303-308.

(2) Li W. Preliminary study on the storage and preservation condition of Tartary buckwheat sprout. Chengdu University Dissertation, 2018.

(3) Zhong L, Niu B, Tang L, Chen F, Zhao G, Zhao J. Effects of polysaccharide elicitor from endophytic Fusarium oxysporum Fat9 on the growth , flavonoid accumulation and antioxidant property of Fagopyrum tataricum sprout cultures. Molecules, 2016, 21: 1590.

(4) Zhong L. Bioactivity evaluation of flavonoid constituents and the effects of endophytic fungi on the growth and flavonoids accumulation of tartary buckwheat sprout cultures. Sichuan University Dissertation, 2016.

(5) Zhao J, Zou L, Zhong L, Peng L, Ying P, Tan M, Zhao G. Effects of polysaccharide elicitors from endophytic Bionectria pityrodes Fat6 on the growth and flavonoid production in tartary buckwheat sprout cultures. Cereal Research Communication, 2015, 43, 661-671.

  1. The description of the extraction is insufficient. Was 70 % methanol (line 254) used? How much was raw material and how much solvent? Were the extracts evaporated to dryness (line 256)?

Response: Thanks very much. The extraction solvent is 70% methanol-water solution, and the ratio of material to liquid is 1:70 (m/v). We have corrected it to make it more accurate.

  1. Key words need improvement.

Response: Thanks. We have improved the key words to make it more informative to the readers.

Specific comments

  1. lines 50, 247 –  How was seed health tested?

Response: Thanks. In our previous investigations, we found that the germination rate and nutritional quality of Tartary buckwheat seeds decreased gradually, with the extension of storage period. For example, the germination rate of new harvested Tartary buckwheat seeds is above 95%. However, the germination rate of Tartary buckwheat seeds could drop to less than 80%, after being placed at room temperature for one year. Furthermore, the germination rate of Tartary buckwheat seeds would decrease to lower than 50%, or even couldn’t germinate after being placed at room temperature for over two years. Therefore, it is necessary to select the good shape and check the germination ability of Tartary buckwheat seeds (according to our experience, the germination rate is usually,higher than 90%, then it could be designated as the health buckwheat seed), for the better growth of buckwheat sprouts.

  1. lines 82, 253 – What type of dryer is it?

Response: Thanks. The type of dryer is DGG-9246A (Qixin, Shanghai, China), and we have added this information in the Experimental section.

  1. line 248 – sodium hypochlorite?

Response: Thanks. Usually, the sodium hypochlorite solution (0.5%-2.0%) has been widely used as a sanitiser in the plant seed and tissue culture field. No changed.

  1. line 251 – Which lamps, which light intensity?

Response: Thanks very much. In this research, the lamp is light-emitting diodes (LEDs), and the light intensity is 7500 lx. We have added this information in the Experimental section to make it more informative to the readers.

  1. line 271 –Is it really about removing MICROBES from samples for chromatographic separation?

Response: Thanks. According to many researchers’ reports and our experience, the filtrate membrane (0.22 μm and 0.45 μm) has been widely used to remove the particles or microbes from samples. We have rephrased this sentence to make it more accurate according to your suggestion.

Round 2

Reviewer 1 Report

In my opinion the manuscript entitled "Chemical Profile, Antimicrobial and Antioxidant Activity Assessment of the Crude Extract and Its Main Flavonoids from Tartary Buckwheat Sprouts" can be published in this form.

Reviewer 2 Report

It has to be said that the article looks better after the corrections and additions. It sufficiently meets the requirements of a paper for publication in a scientific journal.